# The Beta-Lactam Resistome Expressed by Aerobic and Anaerobic Bacteria Isolated from Human Feces of Healthy Donors

**DOI:** 10.3390/ph14060533

**Published:** 2021-06-03

**Authors:** Rosalino Vázquez-López, Sandra Solano-Gálvez, Diego Abelardo Álvarez-Hernández, Jorge Alberto Ascencio-Aragón, Eduardo Gómez-Conde, Celia Piña-Leyva, Manuel Lara-Lozano, Tayde Guerrero-González, Juan Antonio González-Barrios

**Affiliations:** 1Departamento de Microbiología, Centro de Investigación en Ciencias de la Salud (CICSA) Facultad de Ciencias de la Salud Universidad Anáhuac Mexico Norte, Huixquilucan 52786, Mexico; rosalino.vazquez@anahuac.mx (R.V.-L.); diego.alvarez@anahuac.mx (D.A.Á.-H.); jorgeascarg@gmail.com (J.A.A.-A.); 2Departamento de Microbiología y Parasitología, Facultad de Medicina, Universidad Nacional Autónoma de México, Coyoacán, Ciudad de México 04510, Mexico; solano.sandrasg@gmail.com; 3Faculty of Infectious and Tropical Diseases, London School of Hygiene & Tropical Medicine, Bloomsbury, London WC1E 7HT, UK; 4Laboratorio de Investigación en Inmunobiología, Facultad de Medicina, Benemérita Universidad Autónoma de Puebla (BUAP), Puebla 72420, Mexico; eduardo.gomezc@imss.gob.mx; 5Laboratorio de Medicina Genómica, Hospital Regional “Primero de Octubre”, ISSSTE, Av. Instituto Politécnico Nacional 1669, Lindavista, Gustavo A. Madero, Ciudad de México 07300, Mexico; plceliaqfb62812@gmail.com (C.P.-L.); manuellara.mvz@gmail.com (M.L.-L.); taygercita.19@gmail.com (T.G.-G.)

**Keywords:** microbiome, beta-lactams, beta-lactamases, resistome

## Abstract

Antibiotic resistance is a major health problem worldwide, causing more deaths than diabetes and cancer. The dissemination of vertical and horizontal antibiotic resistance genes has been conducted for a selection of pan-resistant bacteria. Here, we test if the aerobic and anaerobic bacteria from human feces samples in health conditions are carriers of beta-lactamases genes. The samples were cultured in a brain–heart infusion medium and subcultured in blood agar in aerobic and anaerobic conditions for 24 h at 37 °C. The grown colonies were identified by their biochemical profiles. The DNA was extracted and purified by bacterial lysis using thermal shock and were used in the endpoint PCR and next generation sequencing to identify beta-lactamase genes expression (OXA, VIM, SHV, TEM, IMP, ROB, KPC, CMY, DHA, P, CFX, LAP, and BIL). The aerobic bacterias Aeromonas hydrophila, Citrobacter freundii, Proteus mirabilis, Providencia rettgeri, Serratia fonticola, Serratia liquefaciens, Enterobacter aerogenes, Escherichia coli, Klebsiella pneumoniae, Pantoea agglomerans, Enterococcus faecalis, and Enterobacter cloacae, the anaerobic bacteria: Capnocytophaga species, Bacteroides distasonis, Bifidobacterium adolescentis, Bacteroides ovatus, Bacteroides fragilis, Eubacterium species, Eubacterium aerofaciens, Peptostreptococcus anaerobius, Fusobacterium species, Bacteroides species, and Bacteroides vulgatus were isolated and identified. The results showed 49 strains resistant to beta-lactam with the expression of blaSHV (10.2%), blaTEM (100%), blaKPC (10.2%), blaCYM (14.3%), blaP (2%), blaCFX (8.2%), and blaBIL (6.1%). These data support the idea that the human enteric microbiota constitutes an important reservoir of genes for resistance to beta-lactamases and that such genes could be transferred to pathogenic bacteria.

## 1. Introduction

The indiscriminate and unconscious use of antibiotics in the clinic, agriculture, and livestock have been increased the selection pressure in the worldwide microbiome, inducing resistant, multiresistant, and pan-resistant pathogen bacteria selection [1,2,3]. This problem reached a maximal level during the last two years where the antibiotic use has increased drastically due to the pandemic spread of SARS-CoV-2, since more than 70% of COVID-19 patients have received complimentary treatment with antibiotics drugs [4,5].

Bacterial resistance to antibiotics is a situation that has been increasing during the last decades, converting into a wide world public health problem [6,7], that complicates the treatment of the infection while increases the mortality rate in both nosocomial or community-acquired infectious diseases [8,9]. This resistance is reaching a critical point, as the increase in these resistant, multi-resistant, and pan-resistant pathogenic strains causes serious medical complications [10].

The lactam groups of molecules are integrated into four families, i.e., beta-lactam [11], gamma-lactam [12], delta-lactam [13], and epsilon-lactam [14], with a specific action and not necessarily like antibiotics. Specifically, the beta-lactams drugs show antibiotic effects in a wide bacterial variety. They are characterized by a beta-lactam ring presence, which confers the antimicrobial effect by the transpeptidases and carboxypeptidases inactivation (the proteins responsible for the biosynthesis of the cell wall) through their binding to Penicillin Binding Protein (PBPs) receptors [15]. The beta-lactam family is made up of various members, such as penicillins B and G, amoxicillin, ampicillin, cephalosporins (cephalothin, cephalexin, ceftriaxone, and cefepime), carbapenems (imipenem, meropenem, ertapenem), aztreonbactam, monobactam, and some others. The indiscriminate and unconscious use of these antibiotics in daily clinical practice has caused an increase in multiresistant Enterobacteriaceae strains, especially Escherichia coli and Klebsiella pneumoniae, turning this situation into a serious global health problem [16]. During the last decade that there has been a significant increase in the presence of multi-resistant strains worldwide, mainly in the European Union. The European Read for Antimicrobial Resistance Surveillance (EARSNet) has been identified pan-resistant *Escherichia coli* and *Klebsiella pneumoniae* strains [17,18]. The bacteria have various resistance mechanisms, mainly associated with plasmid assimilation, which allow cross-resistance against a wide range of antibiotics [19]. The most widely described mechanism of antibiotic resistance that is mediated by plasmid transfer is the enzymatic degradation of the beta-lactam ring through the expression of beta-lactamases [20]. The genes encoding for the beta-lactamases enzymes have been widely spread among the bacterial population, through mobile genetic elements, which confer a plasmid-mediated multi-resistance [21,22]. These multi-resistance bacteria complicate the patient treatments and condition the type of antibiotics that can be administrated for the treatment of common infections [23]. The first-line antibiotic treatment regimens for gastrointestinal and airway infections usually include beta-lactams drugs, such as ampicillin, amoxicillin, or third generation cephalosporins (Ceftriaxone), but the resistance determines the treatment that can be given to patients and decreases the effectiveness significantly of these treatments. The Carbapenems (ertapenem, meropenem, imipenem) have been widely used as an alternative for the treatment of these resistant bacteria, however, in recent years, resistant bacteria strains to these antibiotics have been identified [24]. The main causes associated with this increase in bacterial resistance are the error in the medical prescription, self-medication, incomplete antibiotics treatment, and the use of these drugs by the agricultural and livestock industry for the treatment of crops for human consumption [25].

For these reasons, we test the hypothesis that the Enterobacteriaceae integrating the human gastrointestinal microbiome are carriers of beta-lactamases genes in healthy individuals who have not received antibiotic treatment during their last year of life.

## 2. Results

### 2.1. Bacteria Identification

From the sampling of 20 different medicine students in healthy conditions from Mexico City, 49 beta-lactam resistant strains were isolated, founding 27 strands of 12 different aerobic and 22 strand of 11 different anaerobic beta-lactam resistant Enterobacteriaceae species (Figure 1). The distribution of the identified bacteria was as follows. Aerobic: *Aeromonas hydrophila* 3.7% (*n* = 1), *Citrobacter freundii* 3.7% (*n* = 1), *Enterobacter aerogenes* 7.4% (*n* = 2), *Enterococcus faecalis* 14.8% (*n* = 4), *Enterobacter cloacae* 22.2% (*n* = 6), *Escherichia coli* 11.1% (*n* = 3), *Klebsiella pneumoniae* 11.1%% (*n* = 3), *Proteus mirabilis* 3.7% (*n* = 1), *Providencia rettgeri* 3.7% (*n* = 1), *Pantoea agglomerans* 11.1% (*n* = 3), *Serratia fonticola* 3.7% (*n* = 1), *Serratia liquefaciens* 3.7% (*n* = 1), Anaerobic: *Bacteroides distasonis* 4.5% (*n* = 1), *Bifidobacterium adolescentis* 4.5% (*n* = 1), *Bacteroides ovatus* 4.5% (*n* = 1), *Bacteroides fragilis* 4.5% (*n* = 1), *Bacteroides species* 13.6% (*n* = 3), *Bacteroides vulgatus* 18.2% (*n* = 4), *Capnocytophaga species* 4.5% (*n* = 1), *Eubacterium species* 4.5% (*n* = 1), *Eubacterium aerofaciens* 18.1% (*n* = 4), *Fusobacterium species* 13.6% (*n* = 3) and *Peptostreptococcus anaerobius* 9.1% (*n* = 1) (Figure 1). All these data were confirmed by whole DNA sequencing using NGS (Table 1), showing 98% of concordance with the automatized bacterial identification. All bacteria genomes size sequenced were similar to those reported in the gene bank, observing Δ from 1% to 8% of differences compared with the reported sizes (Table 2). 

### 2.2. Antimicrobial Susceptibility Testing

The antimicrobial susceptibility test of beta lactam resistant bacteria showed, that the 88.9% (*n* = 24) showed resistance to amoxicillin/clavulanic acid, ampicillin, and vancomycin, followed by resistance to cefixime with 81.5% (*n* = 22). Resistance towards cefuroxime and towards cephalothin occurred in both cases in 70.4% (*n* = 19) of the isolated aerobic bacteria, followed by resistance towards cefazolin and cefaclor, both with 66.7% (*n* = 18). 63% (*n* = 17) of the aerobic bacteria isolates showed resistance to piperacillin, followed by resistance to doxycycline with 51.9% (*n* = 14) resistance. The lowest resistance was observed towards piperacillin/tazobactam 18.5% (*n* = 5), ceftizoxime 14.8% (*n* = 4) and towards meropenem and imipenem, both with 11.1% (*n* = 3), While 95.5% (*n* = 21) of the isolated anaerobic bacteria showed resistance towards vancomycin followed by resistance towards cefixime with 81.8% (*n* = 18) and towards piperacillin, cephalothin, cefaclor and cefuroxime, each with 72.7% (*n* = 18). 68.2% (*n* = 15) of this population was resistant to cefazolin, followed by resistance towards doxycycline 63.6% (*n* = 14) and towards Amoxicillin/Clavulanic acid and Ampicillin, both with 54.5% (*n* = 12) and towards Ceftizoxime 41.5% (*n* = 10). The lowest resistance was observed towards Piperacillin/Tazobactam 27.3% (*n* = 6) and to Meropenem and Imipenem, each with 4.5% (*n* = 1), and non-beta-lactam resistance is showed in Table 3.

### 2.3. Beta-Lactamases Gene Family Identification 

The endpoint PCR results showed that 100 % of the obtained bacterial strain carries at least a gene of Beta-Lactamases, identified the gene families transcribed and the phenotype responsible for Beta-Lactam resistance: 49 strains were positive to *bla*TEM (100%), five positive strains for the *bla*SHV (10.2%), five positive strains for *bla*KPC (10.2%), seven positive strains for *bla*CYM (14.3%), one positive strain for blaP (2%), four positive strain for *bla*CFX (8.2%), and three positive strains for blaBIL (6.1%) gene families (Table 4). No members of gene families *bla*OXA, *bla*VIM, *blaI*MP, *bla*ROB, *bla*CTX, *bla*DHA, and *bla*LAP were found in all obtained strains from feces of human and healthy conditions. These data were also confirmed by whole-genome sequencing of isolated strains (Table 3 and Table 4).

The isolated strains from human feces in healthy conditions showed a restricted pattern of resistance to beta-lactam drugs. The individual analysis showed that of the blaTEM was present in all (100% *n* = 49) of isolated strains, from these the 77.5% (*n* = 38) was carrier only for this gene family, the other 25.5% of isolated strains showed a mixed genotype for multiple gene families encoding beta-lactamase and specific for each strain, conformed from two (*bla*TEM + *bla*BIL, *bla*TEM + *bla*CYM, or *bla*TEM + *bla*P) to five (*bla*TEM + *bla*SHV + *bla*KPC + *bla*CYMX + *bla*DHA + *bla*BIL) gene families of beta-lactamases (Table 5).

### 2.4. Beta-Lactamases Genotype

The NGS sequencing showed seven members of the TEM beta-lactamases gene family including *bla*TEM-1, *bla*TEM-2, *bla*TEM-3, *bla*TEM-10, *bla*TEM-12, *bla*TEM-24, and *bla*TEM-52 distributed in the 49 *bla*TEM positive strains, two members of KPC beta-lactamases gene family (*bla*KPC-2 and *bla*KPC-3), only one member of each one of CYM (*bla*CMY-2), BIL (*bla*BIL-1), CFX (*bla*CTX-M-15), and P (*bla*P) beta-lactamases gene families were found (Table 4). 

The mixed gene expression pattern was present some aerobic of Enterobacteriaceae species: *Aeromonas hydrophila* (*bla*TEM-1 + *bla*SHV-12 + *bla*BIL-1), *Enterobacter aerogenes* (*bla*TEM-1 + *bla*KPC-2), *Enterococcus faecalis* (*bla*TEM-1 + *bla*SHV-1), *Enterobacter cloacae* (*bla*TEM-12 + *bla*SHV-1), *Escherichia coli* (*bla*TEM-52 + *bla*SHV-12 + *bla*CMY-2 + *bla*CTX-M-15), *Klebsiella pneumoniae* (*bla*TEM-10 + *bla*SHV-12 + *bla*KPC-3 + *bla*CMY-2 + *bla*CTX-M-15), *Proteus mirabilis* (*bla*TEM-1 + *bla*KPC-2 + *bla*CMY-2 + *bla*CTX-M-15), *Pantoea agglomerans* (*bla*TEM-1 + *bla*CMY-2), *Serratia fonticola* (*bla*TEM-1 + *bla*KPC-2), while the anaerobic bacteria that express mixed genotype were: *Bacteroides distasonis* (*bla*TEM-1 + *bla*BIL-1), *Bacteroides fragilis* (*bla*TEM-2 + *bla*CMY-2), *Bacteroides vulgatus* (*bla*TEM-1 + *bla*CMY-2), and *Fusobacterium* species (*bla*TEM-12 + *bla*P). The NGS analyses for all the bacterial strains tested were negative to *bla*OXA, *bla*VIM, *bla*IMP, *bla*ROB, *bla*CTX, *bla*DHA, and *bla*LAP (Table 6).

## 3. Discussion

The wide use of antibiotics in the clinic, agriculture, and livestock induce a high selection pressure in the worldwide microbiome allowing the resistant, multi-resistant, and pan-resistant bacteria selection and colonization of the gut of human and some other animals including insects and seafoodn [1,2,3]. In the same way, contamination affects vegetables for human consumption [26,27,28,29,30,31,32,33,34,35,36,37]. Even though many of these bacteria do not produce pathology, they are capable of transferring these antibiotic resistances mechanisms to human gut microbiota members and pathogenic bacteria, making it a reservoir for antibiotic resistance genes [37]. Following this idea, our research group characterized the beta-Lactam resistome carried by the aerobic and anaerobic Enterobacteriaceae from healthy medical students of Mexico City. 

### 3.1. Enterobacteriaceae Characterization

The data analysis demonstrates the presence of multiple members of aerobic and anaerobic beta-lactam resistant Enterobacteriaceae, the identified bacteria distribution was as follows. Aerobic: *Aeromonas hydrophila*, *Citrobacter freundii*, *Proteus mirabilis*, *Providencia rettgeri*, *Serratia fonticola,* and *Serratia liquefaciens.* Each one represents 3.7% of all beta-lactam resistant aerobic Enterobacteriaceae, the *Enterobacter aerogenes* represent 7.4%, *Enterococcus faecalis* was present in 14.8%, *Enterobacter cloacae* were in the 22.2% of all isolated strands, the more abundant Enterobacteriaceae were *Escherichia coli*, *Klebsiella pneumoniae* in and *Pantoea agglomerans* were found in the 11.1% each one (Figure 1, Table 1 and Table 2). The founded species from human feces are in concordance with previous results were reported the repertoire and variations of human gut microbiota, they characterized 113 different bacteria, including Gram-positive bacteria (*Bifidobacterium*, *Eubacterium*, *Peptostreptococcus*, *Ruminococcus*, *Lactobacillus*, and *Clostridium* genera) and Gram-negative bacteria (Bacteroides, Fusobacteria genera) [38]. All our bacterial culture data were corroborated by NGS sequencing show 98% of concordance in the final bacterial identification. These data are supported by previously published works. Our NGS data are in concordance with other NGS studies that found 71% of all bacterial community was conformed by *Firmicutes*, 9% was *Actinobacteria,* while *Bacteroidetes*, *Proteobacteria,* and *Cyanobacteria* represent a relatively low abundance ranging from 3% to 7%, the bacterial families found were: *Lactobacillacea*, *Ruminococcacea, Lancnospiraceacea, Clostridiaceacea. Streptococceacea, Streptophyta, Staphylococceacea, Verrucomicrobiaceace,* and *Enterobacteriaceae* among others [39]. However, the composition of the intestinal microbiota varies depending on the type of diet [40]. The obtained results from feces samples reveal the potentially pathogenic enterobacteria presence in the human digestive tract in healthy conditions, their presence as the gastrointestinal resident microbiome, and not necessarily associated with an active gut infection. To determine the potential of these batteries to cause infection, it would be necessary to study other factors, such as virulence and pathogenesis genes expressed by each bacterium, bacterial load, and subjects’ immunological status.

### 3.2. Beta-Lactam Gene Families

Due to the isolated Enterobacteriaceae being isolated in presence of the beta-Lactam drug in the culture media (Table 2), all strains were expected to be carriers at least one member of the different gene families that code for beta-Lactamases. The endpoint PCR identified the gene families transcribed and the phenotype responsible for Beta-Lactam resistance: 100% of the isolated strain were positive to blaTEM, 10.2% positive for the blaSHV, 10.2% positive to blaKPC, 10.2% positive for blaCYM (14.3%), 2% positive to blaP, 8.2% positive to blaCFX and 6.1 positive blaBIL gene family (Table 3 and Table 4), no strand positives to members of blaOXA, blaVIM, blaIMP, blaROB, blaCTX, blaDHA, and blaLAP gene families were found (Table 3 and Table 4). These results accord with previous works, finding that Enterobacteriaceae expressed a member of *bla*CTX, *bla*SHV, and *bla*TEM genes families [41], while around the world the main reported beta-lactamase families in the multidrug-resistant Enterobacteriaceae are: *bla*KPC, *bla*TEM, *bla*OKP, *bla*OXA, *bla*SHV, *bla*VIM and *bla*NDM [42], in agreement with our results.

### 3.3. Beta-Lactam Phenotypes

The sequencing data shows that 83.7% of all resistant Enterobacteriaceae was a carrier of *bla*TEM gene family including *bla*TEM-1, *bla*TEM-2, *bla*TEM-10, *bla*TEM-12, *bla*TEM-24, and *bla*TEM-52 members, like a unique or multiple beta-lactam resistance mechanisms, for the *bla*SHV and *bla*KPC gene family we found only two members of each family including *bla*SHV-1, blaSHV-12, blaKPC-2 and blaKPC-3, while for families *bla*CMY, *bla*BIL, *bla*CTX, and *bla*P we only find a member of each one of them including *bla*CMY-2, *bla*BIL-1, *bla*CTX-M-15, and *bla*P-1, all of these beta-lactamases genes were found in combination with a member of the *bla*TEM gene family. In specific, we found that blaTEM-1 a non-xtended-spectrum beta-lactamase (ESBL) was dominant in unique beta-lactamase gene expression, while the 16.3% of all resistant Enterobacteriaceae was carrier of mixed phenotype with an ESBL, in specific in *Aeromonas hydrophila (bla*TEM-1 + *bla*SHV-12 + *bla*BIL-1), *Enterobacter aerogenes* (*bla*TEM-1 + *bla*KPC-2), *Enterococcus faecalis* (*bla*TEM-1 + *bla*SHV-1), *Enterobacter cloacae* (*bla*TEM-12 + *bla*SHV-1), *Escherichia coli* (*bla*TEM-52 + *bla*SHV-12 + *bla*CMY-2 + *bla*CTX-M-15), *Klebsiella pneumoniae* (*bla*TEM-10 + *bla*SHV-12 + *bla*KPC-3 + *bla*CMY-2 + *bla*CTX-M-15), *Proteus mirabilis* (*bla*TEM-1 + *bla*KPC-2 + *bla*CMY-2 + *bla*CTX-M-15), *Pantoea agglomerans* (*bla*TEM-1 + *bla*CMY-2), *Serratia fonticola* (*bla*TEM-1 + *bla*KPC-2), while the anaerobic bacteria that express mixed genotype were: *Bacteroides distasonis* (*bla*TEM-1 + *bla*BIL-1), *Bacteroides fragilis* (*bla*TEM-2 + *bla*CMY-2), *Bacteroides vulgatus* (*bla*TEM-1 + *bla*CMY-2) and *Fusobacterium species* (*bla*TEM-12 + *bla*P), in the NGS analysis, all bacterial strains tested were negative to blaOXA, blaVIM, blaIMP, blaROB, blaDHA and blaLAP (Table 5). These results are in agreement with a Korean report were they found that the *bla*TEM-1 is the most prevalent beta-lactamase expressed by the stool Enterobacteriaceae of healthy human [43]. In the same way, the Swiss study conducted with Enterobacteriaceae isolated from healthy humans found that the *bla*CTX-M-14 and *bla*CTX-M-15 are the most ESBL prevalent and are generally expressed as unique genes. Here, too, it is reported that *bla*TEM-1 + *bla*CTX-M1, *bla*TEM-1 + *bla*CTX-M-14 and *bla*TEM-1 + *bla*CTX-M-14 phenotypes are the most prevalent [44], in agreement with our results, which found that *bla*CTX-M-15 is always expressed in mixed genotype.

The bacterial isolates obtained were identified with simple or complex beta-lactam resistome. This is especially relevant since the resistance gene can be transmitted from among a wide range of Enterobacteriaceae through plasmid and mobile genetic elements, conferring antibiotic resistance to pathogenic bacteria that are causing gastrointestinal and systemic infection, complicating their treatment. These findings are especially important since the complex resistome expressed by the Enterobacteriaceae were integrated by one member of *bla*TEM gene family, including *bla*TEM-1, *bla*TEM-2, *bla*TEM-10, *bla*TEM-12, *bla*TEM-24, and *bla*TEM-52, unique or in combination one or more member of *bla*SHV, *bla*KPC, *bla*CMY, *bla*BIL, *bla*CTX, and *bla*P gen families, including *bla*SHV-1, *bla*SHV-12, *bla*KPC-2, *bla*KPC-3, *bla*CMY-2, *bla*BIL-1, *bla*CTX-M-15, or *bla*P-1 (Table 3 and Table 4), which confer resistance to ampicillin, penicillins including oxacillin, extended-spectrum cephalosporins (cefotaxime, ceftazidime), cephamycin, carbapenems, and monobactams characteristics that limit the therapeutic option of infections caused by bacterial carriers of these genes.

The data analysis showed multiple members of aerobic and anaerobic Enterobacteriaceae that show multi-resistance molecular mechanisms to beta-lactams drugs, mediated by unique resistance genes or conforming a beta-lactam resistome, in asymptomatic individuals. This can be widely explained by mechanisms such as antibiotics self-medication, abuse, and error in their medical prescription, as well as lack of adherence to antibiotic treatment, including via the consumption of contaminated water [45,46,47] or foods, mainly meats [48,49], fruits [50], and vegetables [26,51]. Moreover, the agriculture [52] and livestock industries [53,54] indiscriminately use the beta-lactam drugs to maximize production [55,56].

## 4. Materials and Methods

### 4.1. Bioinformatic Analysis and Primer Design

The bioinformatic analysis and primer design were previously described and reported by our research group [26]. Briefly, we obtained the beta-Lactamases integrons and DNA sequences from the GenBank of NCBI, all reported sequences of beta-lactamases we found were used to design primer for the PCR. The DNA sequence alignments were made with ClustalW v.2 software (http://www.clustal.org/clustal2/ accessed on 10 April 2021) [27,28,29], to obtain the phylogenetic trees by using FigTree V1.4.0 software (http://tree.bio.ed.ac.uk/software/figtree/ accessed 10 April 2021) [30]. The conserved sequences found in the alignment were used to design a set of specific and degenerate primers (PerPrimer v1.1.21 Software, http://perlprimer.sourceforge.net accessed 10 April 2021 [31] under astringent criteria, length (18–25 bp), Tm (60–62 °C), GC (40–60 %), ∆T° (1 °C), Amplicons (83–230 bp) [26]. 

### 4.2. Sampling

Twenty volunteers (10 men and 10 women) were selected for this study. All of them were medicine students from Mexico City enrolled at the Anahuac University, aged between 20 and 22 years, with no history of beta-lactam antibiotics treatment in the last year or have been suffering diarrhea during that period. Donors were provided with the sample collection material, which consisted of plastic containers, gloves, a cardboard paper sheet, and wooden spatulas; all the material was sterile. The donors were asked to deposit a portion of their stool on the cardboard sheet and from this, they collect with the wooden spatula, an approximate volume of 10 cubic cm, and place it in the sterile container. The time elapsed from when the sample was taken to its beginning of processing in the laboratory was not more than 2 h, this in order not to alter, as far as possible, the bacterial viability. Then, five g of each human feces samples were weighed and cultured in BHI (brain and heart infusion broth) (Becton Dickinson, Franklin Lakes, NJ) incubated at 37 °C for 24 h, in both aerobic and anaerobic conditions.

### 4.3. Bacterial Isolation

For the anaerobic conditions, the BHI tubes containing samples were immediately placed in the BD BBL ™ GasPak ™ anaerobic jar, one envelope of BD BBL ™ CO2 gas generators and another of BD BBL ™ GasPak ™ anaerobic indicator were placed on them. Later they were incubated at 37 °C during 24 h. For the aerobic conditions, the tubes with BHI containing the samples were directly incubated at 37 °C for 24 h. After is incubation, 10 μL from the BHI were taken, subcultured in MacConkey agar (Becton Dickinson, Franklin Lakes, NJ, USA), and incubated at 37 °C for 24 h. The obtained colonies were subcultured in blood agar and incubated at 37 °C for 24 h, in both aerobic or anaerobic conditions. 

### 4.4. Biochemical Bacterial Identification

The automatized bacterial identification was made according to previously reported by our research group. Briefly, bacterial suspensions were made, by depositing two or three medium-sized colonies (2 to 3 mm) in BBL™ Crystal™ Inoculum Broth (Becton Dickinson, Franklin Lakes, NJ, USA). Said inoculum was adjusted to a Mac Farland 1.0 scale (Expected CFU/mL 3.0 × 10^8^). The inoculum was deposited in BBL™ Crystal™ Enteric/Nonfermenter or Anaerobe ID Kit plates, incubated at 37 °C for 18 h, without CO_2_ and 40–60% humidity. Finally, the plates were read by the BBL™ Crystal™ AutoReader (Becton Dickinson, Franklin Lakes, NJ, USA) and the results were analyzed with the BBL™ Crystal™ MIND v.5.05 Software (Becton Dickinson, Franklin Lakes, NJ, USA) [26].

### 4.5. Antimicrobial Susceptibility Testing

The antimicrobial susceptibility testing was done following a previously reported method [26]. Briefly, the pure colonies obtained from blood agar were resuspended in bacterial suspensions, and were made depositing two or three medium-sized colonies (2 to 3 mm) in BBL Crystal Inoculum Broth (Becton Dickinson; Franklin Lakes, NJ, USA). The obtained inoculum was adjusted while using a Mac Farland 0.5 reading (Expected CFU/mL 1.5 × 108), and cultured in Müeller Hinton 150 × 15 mm^2^ media BD BBL (Becton Dickinson Franklin Lakes, NJ, USA). The antibiotic discs were applied with the Sensi-Disc Designer Dispenser System. The antibiotics panel was conformed by ampicillin (10 µg), ampicillin/sulbactam (10/10 µg), mezlocillin (75 µg), carbenicillin (100 µg), piperacillin/tazobactam (100/10 µg), cefazolin (30 µg), cefaclor (30 µg), cefepime (30 µg), cefoperazone (75 µg), and cefotetan (30 µg) from Becton Dickinson (Franklin Lakes, NJ, USA), Müeller Hinton medium were incubated at 37 °C for 24 h in both aerobic and anaerobic conditions. In anaerobic conditions, the media were placed in an anaerobic jar (BD BBL™ GasPak™), with a C02 gas generators envelope (BD BBL™) and another envelope of BD BBL™ GasPak™ anaerobic indicator placed on them.

### 4.6. DNA Extraction

#### 4.6.1. Crude Extract

Five mL of Luria-Bertani broth (LB) were inoculated with an isolated bacterial colony and incubated overnight at 37 °C with continuous shaking (200 rpm), the bacterial culture was centrifuged at 4000 rpm for five minutes at room temperature, bacterial the pellet was resuspended on 1 mL of sterile free-RNAse and free-DNAse deionized water, the bacterial suspension was heated at 94 °C for 10 min followed by ice shock. Finally, the sample of lysed bacterial was kept at –80 °C until use [26].

#### 4.6.2. Genomic DNA Extraction

The DNA extraction was done following a previously reported method [26]. Briefly, 500 µL of bacterial culture was obtained from a tested single bacterial colony for genomic DNA isolation by RTP pathogen kit (Invitek, Germany) following the manufacturer’s instruction. The eluted DNA solution was quantified by absorbance and its integrity was verify by 2% agarose gel electrophoresis. The sample was kept at –20 °C until use.

#### 4.6.3. Plasmid DNA Extraction

The plasmid DNA extraction was done following a previously reported method [26], by using plasmid DNA isolation by using PureLink HiPure Plasmid DNA Purification Kit (Invitrogen, USA) following the manufacturer’s instructions. Plasmid DNA solution was quantified by digital spectrophotometry by using a NanoDrop spectrophotometer (Thermo Fisher Scientific, Wilmington, DE, USA) and its integrity was verified by electrophoresis in 2 % agarose gel, the sample was kept at –20 °C until use.

### 4.7. Endpoint PCR

The endpoint PCR to identify the β-Lactamases gene families was made by using previously reported specific and degenerated primes (Table 7) [26]: Amplification was made in 25 μL of the reaction mixture containing 2.5 μL of 10x PCR buffer (100 mM TRIS•HCI, 15 mM MgCl2, and 500 mM KCl, pH 8.3), 200 nM each dNTP, 10 μM each primer, 1 U Taq DNA polymerase (Invitrogen; Carlsbad, CA, USA), and 2 μL of crude extract or 10 ng of DNA (genomic or plasmidic). The PCR conditions were 94 °C for 5 min, then 35 cycles of 94 °C for 30 s, 60 °C for 30 s, 72 °C for 30 s, and finally 72 °C for 10 min. The PCR products were analyzed by electrophoresis in a 2% Agarose gel prestained with ethidium bromide and the image was digitized in a GelLogistic 3000 photodocumenter.

### 4.8. Whole and Plasmid DNA Sequencing

The DNA sequencing was performed following a previously reported method by our research group [26], the indexed libraries that were prepared using a standard Illumina Nextera XT DNA Sample Preparation Kit (FC-131-1096) for small genomes and were sequenced on the MiSeq platform (Illumina; San Diego, CA, USA). Adapters and barcodes were trimmed by the default setting in the Illumina experiment manager, generating 300-bp paired-end reads. The quality of the unprocessed reads was assessed using FastQC High Throughput Sequence QC Report v:0.11.5 (Babraham Bioinformatics, Babraham Institute; Cambridge, UK) [32]. A minimum Q score of more than 30 for at least 85% of all reads was attained. All reads were mapped using BWA-MEM aligner version 0.7.7-r441 (Wellome trust, Sanger Institute, Hinxton, UK) [33] and SAMtools version 1.3.1 (Wellome trust, Sanger Institute, Hinxton, UK). The NOVO genome assembly was done using the SPAdes Genome Assembler software version 3.11 (CAB, St. Petersburg State University, St. Petersburg, Russia) [34]. The metagenomic analysis for the taxonomic classification of bacteria was done by using the software Kraken taxonomic sequence classification system Version 0.10.5-beta (CCB, Johns Hopkins University, Baltimore, MD, USA) [35]. The beta-lactamase genes were identified by the comparative analysis while using the Basic Local Alignment Search Tool (BLAST, NCBI-NIH, Bethesda, MD, USA) [36].

## 5. Conclusions

The human gut bacteriome is an important reservoir and mediator for the environmental beta-lactam resistome. The cumulative beta-lactam resistome in the Enterobacteriaceae is indicative of the indiscriminate and irrational use of beta-lactam drugs in practically all economical worldwide activities, especially in Mexico City. The gut bacteriome express beta-lactam resistome in healthy conditions is integrated by multiple beta-lactamases gen families, such as *bla*TEM, *bla*SHV, *bla*KPC, *bla*CYM, *bla*BIL, *bla*CFX, *bla*CYM, *bla*BIL, and *bla*P, which altogether confer resistant, multi-resistant, and pan-resistant characteristics against beta-lactam antibiotics.

## Figures and Tables

**Figure 1 pharmaceuticals-14-00533-f001:**
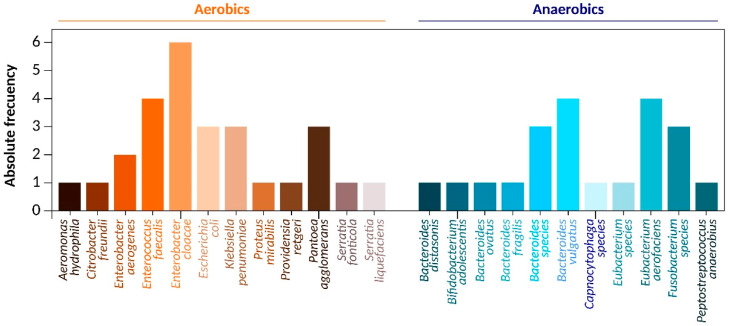
Bacterial frequency. The graph shows the frequency of beta-Lactam resistant aerobic and anaerobic Enterobacteriaceae isolated from feces sample of medicine student in healthy conditions.

**Table 1 pharmaceuticals-14-00533-t001:** Metagenomic identification of the aerobic and anaerobic beta-lactam resistant bacteria isolated from feces cultures of humans in healthy conditions.

Bacteria	Total Reads	Classified Reads	Domain	Phylum	Class	Order	Family	Genus	Species
**Aerobic**									
*Aeromonas hydrophila*	4,502,897	4,412,839 (98%)	4,103,940 (93%)	3,734,586 (91%)	3,211,744 (86%)	2,697,865(84%)	2,212,249 (82%)	1,791,922 (81%)	1,361,861 (76%)
*Citrobacter freundii*	1,816,430	1,798,266 (99%)	1,744,318 (97%)	1,657,102 (95%)	1,491,392 (90%)	1, 327,339(89%)	1,101,691 (83%)	870,336 (79%)	539,608 (62%)
*Enterobacter aerogenes*	897,648	727,095 (81%)	567,134 (78%)	436,693 (77%)	331,887 (76%)	248,915 (75%)	1817,08 (73%)	130,830 (72%)	91,581 (70%)
*Enterococcus faecalis*	3,134,872	3,040,826 (97%)	2,852,734 (91%)	2,821,385 (90%)	2,695,990 (86%)	2,664,641 (85%)	2,539,246 (81%)	2,507,898 (80%)	2,476,549 (79%)
*Enterobacter cloacae*	945,761	898,473 (95%)	892,798 (94%)	871,046 (92%)	866,317 (91%)	850,239 (89%)	842,673 (89%)	836,998 (85%)	835,107 (88%)
*Escherichia coli*	2,361,827	2,078,408 (88%)	1,936,698 (82%)	1,865,843 (79%)	1,606,042 (68%)	1,511,569 (64%)	1,464,333 (62%)	1,440,714 (61%)	1,393,478 (59%)
*Klebsiella pneumoniae*	1,129,675	1,039,301 (92%)	1,016,708 (90%)	1,005,411 (89%)	994,114 (88%)	915,037 (81%)	881,147 (78%)	869,850 (77%)	835,960 (74%)
*Proteus mirabilis*	816,524	808,359 (99%)	805,909 (98%)	803,460 (97%)	767,533 (94%)	751,202 (92%)	743,037 (91%)	738,954 (90%)	726,706 (89%)
*Providencia rettgeri*	354,869	273,249 (77%)	266,152 (75%)	259,054 (73%)	255,506 (72%)	237,762 (67%)	223,567 (63%)	216,470 (61%)	191,629 (54%)
*Pantoea agglomerans*	497,802	438,066 (88%)	433,088 (87%)	408,198 (82%)	393,264 (79%)	388,286 (78%)	353,439 (71%)	338,505 (68%)	333,527 (67%)
*Serratia fonticola*	1,347,964	1,334,484 (99%)	1,267,086 (94%)	1,253,607 (93%)	1,226,647 (91%)	1,186,208 (88%)	1,145,769 (85%)	1,132,290 (84%)	997,493 (74%)
*Serratia liquefaciens*	256,789	190,024 (74%)	182,320 (71%)	179,752 (70%)	174,617 (68%)	159,209 (62%)	156,641 (61%)	154,073 (60%)	133,530 (52%)
Anaerobic									
*Bacteroides distasonis*	845,168	752,201 (89%)	726,844 (86%)	718,393 (85%)	693,038 (82%)	667,683 (79%)	659,231 (78%)	625,424 (74%)	608,521 (72%)
*Bifidobacterium adolescentis*	748,751	718,801 (96%)	715,806 (95%)	704,575 (94%)	700,082 (93%)	689,600 (92%)	687,353 (91%)	682,861 (91%)	658,901 (88%)
*Bacteroides ovatus*	965,743	905,867 (94%)	902,004 (94%)	898,141 (93%)	893,312 (92%)	888,484 (91%)	830,539 (86%)	743,622 (77%)	637,390 (66%)
*Bacteroides fragilis*	986,241	976,379 (99%)	966,516 (98%)	956,654 (97%)	951,723 (96.4%)	946,791 (96.1%)	927,067 (94.3%)	917,204 (93%)	915,232 (92.6%)
*Bacteroides species*	567,284	486,162 (85.7%)	483,893 (85.3%)	482,759 (85.1%)	481,624 (84.9%)	478,788 (84.4%)	478,220 (84.3%)	477,086 (84.1%)	474,817 (83.7%)
*Bacteroides vulgatus*	928,712	911,995 (98.2%)	911,066 (98.1%)	833,055 (89.7%)	830,269 (89.4%)	828,411 (89.2%)	825,625 (88.9%)	819,124 (88.2%)	816,338 (87.9%)
*Capnocytophaga species*	349,817	319,733 (91%)	315,535 (90.2%)	313,471 (89.6%)	312,911 (89.4%)	312,282 (89.2%)	312,212 (89.1%)	299,793 (85.7%)	295,211 (84.3%)
*Eubacterium species*	567,492	542,522 (95.6%)	541,387 (95.4%)	540,309 (95.2%)	539,685 (95.1%)	537,982 (94.8%)	536,280 (94.5%)	535,712 (94.4%)	534,010 (94.1%)
*Eubacterium aerofaciens*	814,927	620,974 (76.2%)	616,085 (75.5%)	604,676 (74.2%)	599,786 (73.6%)	595,712 (73.1%)	594,082 (72.9%)	581,043 (71.3%)	576,968 (70.8%)
*Fusobacterium species*	237,491	204,717 (86.2%)	203,292 (85.6%)	192,843 (81.2%)	191,418 (80.6%)	189,755 (79.9%)	187,855 (79.1%)	186,193 (78.4%)	185,955 (78.3%)
*Peptostreptococcus anaerobius*	729,358	686,691 (94.1%)	686,399 (94.1%)	683,044 (93.6%)	682,314 (93.5%)	679,251 (93.1%)	677,282 (92.8%)	612,442 (83.9%)	587,498 (80.5%)

**Table 2 pharmaceuticals-14-00533-t002:** Whole-genome sequencing characteristics of aerobic and anaerobic Enterobacteriaceae rods isolated from human feces in healthy conditions.

Bacteria	CDS	Number of Sequence Contigs	Genome Size (bp)		
			Assembled	Reported	Difference
Aerobic					
*Aeromonas hydrophila*	4,428	1,124	5,124,487	4,911,246	213,241
*Citrobacter freundii*	5,064	789	5,343,952	5,297,052	46,900
*Enterobacter aerogenes*	4,545	1,484	5,578,724	5,280,350	298,374
*Enterococcus faecalis*	2,969	951	3,111,017	3,038,914	72,103
*Enterobacter cloacae*	4,545	1,484	4,982,176	4,772,910	209,266
*Escherichia coli*	5,704	1,561	5,689,156	5,615,389	73,767
*Klebsiella pneumoniae*	5,071	583	5,479,173	5,315,120	164,053
*Proteus mirabilis*	3,772	584	4,183,869	4,209,445	25,576
*Providencia rettgeri*	4,497	638	4,954,326	4,780,676	173,650
*Pantoea agglomerans*	4,255	738	4,090,220	4,047,712	42,508
*Serratia fonticola*	5,945	916	6,483,043	6,000,511	482,532
*Serratia liquefaciens*	4,936	495	5,706,987	5,395,544	311,443
Anaerobic					
*Bacteroides distasonis*	3,896	603	4,952,323	4,812,038	140,285
*Bifidobacterium adolescentis*	1,742	418	2,173,720	2,089,645	84,075
*Bacteroides ovatus*	4,803	756	6,425,267	6,472,489	47,222
*Bacteroides fragilis*	4,100	1,025	5,188,967	5,474,541	285,574
*Bacteroides species*	2,401	904	2,658,624	2,628,345	30,279
*Bacteroides vulgatus*	5,055	406	5,311,454	5,681,290	369,836
*Capnocytophaga species*	2,208	502	2,614,527	2,837,214	222,687
*Eubacterium species*	2,197	505	2,628,803	2,450,450	178,353
*Eubacterium aerofaciens*	1,887	1,548	2,128,754	2,264,854	136,100
*Fusobacterium species*	2,113	738	2,159,799	2,185,897	26,098
*Peptostreptococcus anaerobius*	1,793	847	1,989,753	2,106,123	116,370

**Table 3 pharmaceuticals-14-00533-t003:** Whole antibiograms characteristics of beta-lactam resistant bacteria isolated from feces cultures of humans in healthy conditions.

Antibiotic Families	Drugs	Aerobics		Anaerobic	
		Frequency	%	Frequency	%
Beta-Lactam	Amoxicillin/Clavulanic acid (AmC-30)	24	88.9	12	54.5
	Piperacillin (PIP-100)	17	63.0	16	72.7
	Piperacillin/Tazobactam (TZP-110)	5	18.5	6	27.3
	Doxycycline (D-30)	14	51.9	14	63.6
	Ampicillin 10(AM-10)	24	88.9	12	54.5
	Cephalothin 1° (CF-30)	19	70.4	16	72.7
	Cefazolin 1° (CZ-30)	18	66.7	15	68.2
	Cefaclor 2° (CEC-30)	18	66.7	16	72.7
	Cefuroxime 2° (CXM-30)	19	70.4	16	72.7
	Cefixime 3° (CFM-5)	22	81.5	18	81.8
	*Ceftizoxime* 3a (ZOX-30)	4	14.8	10	45.5
	Meropenem (MEM-10)	3	11.1	1	4.5
	Imipenem (IPM-10)	3	11.1	1	4.5

**Table 4 pharmaceuticals-14-00533-t004:** Frequency of beta-lactamase gene families identified by endpoint PCR in roads cultured from gut microbiota in healthy conditions.

Gene Family	Positive Strains (*n*)	%	Gene Family	Positive Strains (*n*)	%
*bla*TEM	49	100.0	*bla*BIL	3	6.1
*blaSHV*	5	10.2	*bla*P	1	2.0
*bla*KPC	5	10.2	*bla*CFX	4	8.2
*bla*CYM	7	14.3			

**Table 5 pharmaceuticals-14-00533-t005:** Beta-Lactamases (bla) gene families carried by the aerobic and anaerobic Enterobacteria of humans in healthy conditions.

Aerobic Bacteria	Beta-Lactamases Families						Anaerobic Bacteria	Beta-Lactamases Families			
	TEM	SHV	KPC	CYM	BIL	CFX		TEM	CYM	BIL	P
*Aeromonas hydrophila*	**+**	**+**			**+**		*Capnocytophaga species*	**+**			
*Citrobacter freundii*	**+**						*Bacteroides distasonis*	**+**		**+**	
*Proteus mirabilis*	**+**		**+**	**+**		**+**	*Bifidobacterium adolescenti*	**+**			
*Providencia rettgeri*	**+**						*Bacteroides ovatus*	**+**			
*Serratia fonticola*	**+**		**+**				*Bacteroides fragilis*	**+**	**+**		
*Serratia liquefaciens*	**+**						*Eubacterium species*	**+**			
*Enterobacter aerogenes*	**+**		***+***				*Eubacterium aerofaciens*	**+**			
*Escherichia coli*	**+**	**+**		**+**		**+**	*Peptostreptocuccus anaerobius*	**+**			
*Klebsiella pneumoniae*	**+**	**+**	**+**	**+**		**+**	*Fusobacterium species*	**+**			**+**
*Pantoea agglomerans*	**+**			**+**			*Bacteroides vulgatus*	**+**	**+**		
*Enterococcus faecalis*	**+**	**+**					*Bacteroides species*	**+**			
*Enterobacter cloacae*	**+**	**+**	**+**	**+**		**+**					

(**+**) Positive for the beta-lactamase gene family.

**Table 6 pharmaceuticals-14-00533-t006:** Mixed Genotype of the beta-lactam antibiotic resistome identified by next-generation sequencing in the aerobic and anaerobic Enterobacteria from feces of humans in healthy conditions.

Bacteria	Phenotype of Beta-Lactamases						
	TEM	SHV	KPC	CYM	BIL	CFX	P
**Aerobic**							
*Aeromonas hydrophila*	*bla*TEM-1	*bla*SHV-12			*bla*BIL-1		
*Citrobacter freundii*	*bla*TEM-2						
*Enterobacter aerogenes*	*bla*TEM-1		*bla*KPC-2				
*Enterococcus faecalis*	*bla*TEM-1	*bla*SHV-1					
*Enterobacter cloacae*	*bla*TEM-12	*blaSHV*-1					
*Escherichia coli*	*bla*TEM-52	*bla*SHV-12		*bla*CMY-2		*bla*CTX-M-15	
*Klebsiella pneumoniae*	*bla*TEM-10	*bla*SHV-12	*bla*KPC-3	*bla*CMY-2		*bla*CTX-M-15	
*Proteus mirabilis*	*bla*TEM-1		*bla*KPC-2	*bla*CMY-2		*bla*CTX-M-15	
*Providencia rettgeri*	*bla*TEM-24						
*Pantoea agglomerans*	*bla*TEM-1			*bla*CMY-2			
*Serratia fonticola*	*bla*TEM-1		*bla*KPC-2				
*Serratia liquefaciens*	*bla*TEM-12						
**Anaerobic**							
*Bacteroides distasonis*	*bla*TEM-1				*bla*BIL-1		
*Bifidobacterium adolescentis*	*bla*TEM-1						
*Bacteroides ovatus*	*bla*TEM-1						
*Bacteroides fragilis*	*bla*TEM-2			*bla*CMY-2			
*Bacteroides species*	*bla*TEM-1						
*Bacteroides vulgatus*	*bla*TEM-1			*bla*CMY-2			
*Capnocytophaga species*	*bla*TEM-1						
*Eubacterium species*	*bla*TEM-1						
*Eubacterium aerofaciens*	*bla*TEM-1						
*Fusobacterium species*	*bla*TEM-12						*bla*P-1
*Peptostreptococcus anaerobius*	*bla*TEM-1						

**Table 7 pharmaceuticals-14-00533-t007:** Primers sequences and amplicon size for beta-lactamase gene family [26].

Gene Family	Primer Name	Primer Sequence (5’ to 3’)	Tm (°C)	Position	Amplicon (bp)
blaOXA	BlaOXA-FW	GGTTTCGGTAATGCTGAAATTGG	61.18	214–236	114
	BlaOXA-RW	GCTGTGTATGTGCTAATTGGGA	61.19	327–306	
blaVIM	BlaVIM-FW	CGACAGTCARCGAAATTCC	61.39	105–123	133
	BlaVIM-RW	CAATGGTCTSATTGTCCGTG	61.34	238–219	
blaSHV	BlaSHV-FW1	CGTAGGCATGATAGAAATGGATC	61.04	133–155	106
	BlaSHV-RW1	CGCAGAGCACTACTTTAAAGG	61.33	239–218	
	BlaSHV-FW2	GCCTCATTCAGTTCCGTTTC	61.62	399–418	141
	BlaSHV-RW2	CCATTACCATGAGCGATAACAG	61.22	540–518	
blaTEM	BlaTEM-FW	GCCAACTTACTTCTGACAACG	61.80	1699–1719	213
	BlaTEM-RW	CGTTTGGAATGGCTTCATTC	60.13	1912–1892	
blaIMP	BlaIMP-FW1	GGAATAGARTGGCTTAAYTCTCG	60.92	319–332	183
	BlaIMP-RW1	CYASTASGTTATCTKGAGTGTG	62.45	502–480	
	BlaIMP-FW2	GGTGGAATAGARTGGCTTAAYTC	61.11	316–339	192
	BlaIMP-RW2	CCAAACCACTACGTTATCTKGAG	61.29	508–485	
blaROB	BlaROB-FW	CCAACATCGTGGAAAGTGTAG	61.27	718–739	126
	BlaROB-RW	GTAAATTGCGTACTCATGATTGC	60.90	844–821	
blaKPC	BlaKPC-FW	GCTAAACTCGAACAGGACTTTG	61.79	100–121	117
	BlaKPC-RW	CTTGAATGAGCTGCACAGTG	61.90	216–197	
blaCTX	BlaCTX-FW1	GATACCGCAGATAATACGCAG	60.79	161–181	116
	BlaCTX-RW1	CGTTTTGCGTTTCACTCTG	60.28	276–258	
	BlaCTX-FW2	GCTGATTCTGGTCACTTACTTC	61.02	789–810	83
	BlaCTX-RW2	CGCCGACGCTAATACATC	60.69	855–872	
	BlaCTX-FW3	CTGCTTAACTACAATCCSATTGC	62.17	314–336	226
	BlaCTX-RW3	GGAATGGCGGTATTKAGC	60.86	539–522	
blaCMY	BlaCMY-FW1	GTTTGAGCTAGGATCGGTTAG	60.25	337–357	123
	BlaCMY-RW1	CTGTTTGCCTGTCAGTTCTG	61.48	460–441	
	BlaCMY-FW2	GAACGAAGGCTACGTAGCT	61.71	213–231	160
	BlaCMY-RW2	CTGAAACGTGATTCGATCATCA	61.08	372–351	
blaDHA	BlaDHA-FW1	GCATATTGATCTGCATATCTCCAC	61.60	399–422	200
	BlaDHA-RW1	GCTGCTGTAACTGTTCTGC	61.62	598–580	
	BlaDHA-FW2	GCGGATCTGCTGAATTTCTATC	61.54	464–485	147
	BlaDHA-RW2	GCAGTCAGCAACTGCTCATAC	61.05	610–591	
	BlaDHA-FW3	GTAAGATTCCGCATCAAGCTG	61.74	430–450	117
	BlaDHA-RW3	GGGTTATCTCACACCTTTATTACTG	61.08	546–522	
blaP	BlaP-FW	GGAGAATATTGGGATTACAATGGC	61.74	271–294	204
	BlaP-RW	CGCATCATCGAGTGTGATTG	61.80	474–455	
blaCFX	BlaCFX-FW	CCAGTCATATCATTGACAGTGAG	60.86	437–459	177
	BlaCFX-RW	GACATTTCCTCTTCCGTATAAGC	61.16	613–591	
blaLAP	BlaLAP-FW	AGGGCTTGAACAACTTGAAC	61.07	249–268	126
	BlaLAP-RW	GTAATGGCAGCATTGCATAAC	60.59	374–354	
blaBIL	BlaBIL-FW	GCCGATATCGTTAATCGCAC	61.65	100–119	128
	BlaBIL-RW	GTTATTGGCGATATCGGCTTTA	60.98	227–206	

## Data Availability

The data presented in this study are available from the corresponding author.

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
