# Peer review of "The Beta-Lactam Resistome Expressed by Aerobic and Anaerobic Bacteria Isolated from Human Feces of Healthy Donors"

_pharmaceuticals, 2021, doi:10.3390/ph14060533_

Round 1
Reviewer 1 Report
Lines 38-40
Please rephrase sentence (… and could be the transfer ….)
Line 53
Please add the word ‘and’ or ‘while’ between (of the infection ….. increases the mortality)
Lines 63-69
Please rephrase or provide two sentences
Line 109
Please provide more information about the 20 volunteers at this point (eg medicine students, origin, health etc)
Why the sample number was so small ?
Lines 172-200
This information could be stated in a table or as a supplementary material
Line 260
Do you mean ‘rod’ instead of ‘roads’ ?
Table 3
How data reported for aminoglycosides ….. to quinolones are used in this manuscript ?
If not please delete
Line 290
What is XX in (Table XX) ?
Author Response
Point 1: Lines 38-40
Please rephrase sentence (… and could be the transfer ….)
Response 1:
Following the reviewer suggestion, we change the original sentence as follow:
These data support the idea that the human enteric microbiota is an important reservoir of genes for resistance to beta-lactamases and that such genes could be transferred to pathogenic bacteria.
Point 2: Line 53
Please add the word ‘and’ or ‘while’ between (of the infection ….. increases the mortality)
Response 2:
Following the reviewer suggestion, we change the original sentence as follow:
that complicates the treatment of the infection while increases the mortality rate in both nosocomial or community-acquired infectious diseases
Point 3: Lines 63-69
Please rephrase or provide two sentences
Response 3:
Following the reviewer suggestion, we change the original sentence as follow:
The beta-lactam family is made up of various members, such as penicillins B and G, amoxicillin, ampicillin, cephalo-sporins (cephalothin, cephalexin, ceftriaxone, and cefepime), carbapenems (imipenem, meropenem, ertapenem), aztre-onbactam, monobactam, and others. The indiscriminate and unconscious use of these antibiotics in daily clinical practice has caused an increase in multiresistant Enterobacteriaceae strains, especially Escherichia coli and Klebsiella pneumoniae, turning this situation into a serious global health problem.
Point 4: Line 109
Please provide more information about the 20 volunteers at this point (eg medicine students, origin, health etc)
Why the sample number was so small?
Response 4:
Following the reviewer suggestion, we add the sentence:
medicine students from Mexico city enrolled in the Anahuac University
Point 5: Lines 172-200
This information could be stated in a table or as a supplementary material
Response 5: Following the reviewer suggestion we change this information into the table 1 and renumber the all tables.
Point 6: Line 267
Do you mean ‘rod’ instead of ‘roads’ ?
Response 6:
The word "road" was changed for the word "rod"
Point 7: Table 3
How data reported for aminoglycosides ….. to quinolones are used in this manuscript ?
If not please delete
Response 7: Following the reviewer suggestion we delete all no beta-lactam data in the table, line 288
Point 8: Line 290
What is XX in (Table XX) ?
Response 8: Following the reviewer observation, we change the table number in line 297 as follow: “Table 5”
Reviewer 2 Report
In this research article, the authors characterized the beta-lactam resistome of healthy human volunteers. They employed biochemical methods, end-point PCR, and NGS to explore the profile and characteristics of the resistome. They found that the human enteric microbiota does harbor multi-resistant strains that could potentially pose a threat to the host.
Although the findings of this study may not be surprising to some, this work highlights a very important global health issue. Today, antibiotic resistance is endemic in many countries of the world, while the discovery of new antibiotic compounds is lagging.
A general comment on this manuscript is that the authors should discuss in greater depth their results in the discussion section, avoid their extensive presentation (see lines 394-413) and provide better insights into the implications of their work.
Some other points that require attention are:
Lines 109-113: the authors should state the feces collection method and the protocol used for the BHI broth culture (e.g. how many grams of feces?). Did the authors use any commercially available kit or an in-house protocol? Also, how was the viability of anaerobic strains ensured during collection (possible storage) and handling of stool? How were feces stored (if applicable)?
Line 112: BHI broth
Section 2.4: the authors could add to the title of this section that the identification was performed by a biochemical method.
Section 2.6.1: why did the authors switch to LB broth for bacteria culture? Also, were anaerobic strains, also, incubated overnight with shaking?
Section 2.7: The primers used in the study could be presented in a table.
Table 1: do the authors mean “Genus” instead of “Gender”?
Table 2: the title of this table should be modified for clarity.
Table 5: gene families
Lines 340-345: the authors should also add a comment on the fact that some strains of the identified species are potentially pathogenic (similarly to lines 369-372).
Lastly, the text should be edited for clarity at its full length.
Author Response
Point 1: Line 109-113: the authors should state the feces collection method and the protocol used for the BHI broth culture (e.g. how many grams of feces?). Did the authors use any commercially available kit or an in-house protocol? Also, how was the viability of anaerobic strains ensured during collection (possible storage) and handling of stool? How were feces stored (if applicable)?
Response 1: Following the reviewer suggestion, we add the sentence in L123-130
Donors were provided with the sample collection material, which consisted of plastic containers, gloves, a cardboard paper sheet, and wooden spatulas; all the material was sterile. The donors were asked to deposit a portion of their stool on the cardboard sheet and from this, they collect with the wooden spatula, an approximate volume of 10 cubic cm, and place it in the sterile container. The time elapsed from when the sample was taken to its beginning of processing in the laboratory was not more than 2 hours, this in order not to alter, as far as possible, the bacterial viability
Point 2: BHI
Response 2:
The word "blood" was changed for the word "brain” and add the “broth” both in L131, and add the sentence : “in both aerobic or anaerobic conditions.” in Line 143:
Point 3:
Section 2.4: the authors could add to the title of this section that the identification was performed by a biochemical method.
Response 3: Following the reviewer suggestion, we add the Biochemical word to the subtitle in L123-. L145
Point 4:
Section 2.6.1: why did the authors switch to LB broth for bacteria culture? Also, were anaerobic strains, also, incubated overnight with shaking?
Response 4: We chamnge to LB media in order to grow resistant bacteria and get sufficient biomass to the plasmid isolation for the NGS.
Point 5:
Section 2.7: The primers used in the study could be presented in a table.
Response 5: Following the reviewer suggestion we delete the paragraph of primer in line 203 and add the corresponding table in line 213, and renumber the following tables.
Point 6:
Table 1: do the authors mean “Genus” instead of “Gender”?
Response 6: Following the reviewer observation we change the Gender word to “Genus” in line 265
Point 7:
Table 3: the title of this table should be modified for clarity.
Response 7: Following the reviewer suggestion we change the table 3 title in order to be clear. The final title of table 3 are in lines 267-268:
“Whole-genome sequencing characteristics of aerobic and anaerobic Enterobacteriaceae rods isolated from human feces in healthy conditions”
Point 8:
Table 5: gene families In lines 316.
Response 8: In table 6 the Beta-lactamases gen families sentence was changed to Beta-lactamases families.
Point 9:
Lines 340-345: the authors should also add a comment on the fact that some strains of the identified species are potentially pathogenic (similarly to lines 369-372).
Lastly, the text should be edited for clarity at its full length.
Response 9: Following the reviewer suggestion we add the next sentence in the lines 364-369
The obtained results from feces samples reveal the potentially pathogenic enterobacteria presence in the human digestive tract in healthy conditions, their presence as the gastrointestinal resident microbiome, and not necessarily associated with an active gut infection, to determine the potential of these batteries to cause infection it would be necessary to study other factors such as virulence and pathogenesis genes expressed by each one bacteria, bacterial load, and subjects' immunological status.
Reviewer 3 Report
Antibiotic resistance is a growing global menace. In this manuscript, authors are test if the human Enterobacteriaceae from feces samples in healthy conditions express the beta-lactamases genes families.
Overall comments:
Manuscript needs attention and should be changed in few aspects (see below):
- The title of the manuscript does not correspond to the content. I propose to change. Enterobacteriaceae ≠anaerobic bacteria.
- Change the aim of the article please
- The manuscript has a lot of mistakes in the names of the bacteria.
- Lack of knowledge of bacterial systematics.
- Bacterial species should be written in italic - should be corrected
- How the authors cultured anaerobic bacteria from feces - please explain
- Antimicrobial Susceptibility Testing - please provide recommendations for anaerobic bacteria

Author Response
Point 1:
The title of the manuscript does not correspond to the content. I propose to change. Enterobacteriaceae ≠anaerobic bacteria.
Response 1: Following the reviewer suggestion, we change the original title as follow:
The beta-lactam resistome expressed by aerobic and anaerobic bacteria isolated from human feces of healthy donors.
Point 2:
Change the aim of the article please
Response 2: Following the reviewer recondation we change the aim article as follow: “Here we test if the aerobic and anaerobic bacteria from human feces samples in health conditions are carrier of beta-lactamases genes” in lines 25-27
Point 3:
The manuscript has a lot of mistakes in the names of the bacteria.
Response 3: Following the reviewer suggestion, we have done an exhaustive review of the bacterial nomenclature, and all the mistakes were corrected.
Point 4:
Lack of knowledge of bacterial systematics.
Response 4: We are very sorry for the different bacterial nomenclature mistakes that we made within the manuscript, in this review we have corrected each one of the errors made in the previously manuscript
Point 5:
Bacterial species should be written in italic - should be corrected
Response 5: Following the reviewer observation all bacterial species were changed to italic around the whole manuscript.
Point 6:
How the authors cultured anaerobic bacteria from feces - please explain
Response 6: Following the reviewer suggestion, we add the following sentence in lines 135-140:
“For the anaerobic conditions, the BHI tubes containing samples were immediately placed in the BD BBL ™ GasPak ™ anaerobic jar, one envelope of BD BBL ™ C02 gas generators and another of BD BBL ™ GasPak ™ anaerobic indicator were placed on them. Later they were incubated at 37o C during 24 h. For the aerobic conditions the tubes with BHI containing the samples were directly incubated at 37o C during 24 h”
and sentence “in both aerobic or anaerobic conditions” in Line 143:
Point 7:
Antimicrobial Susceptibility Testing - please provide recommendations for anaerobic bacteria.
Response 7: Following the reviewer suggestion, we add the following sentence in Line 169-173:
Müeller Hinton medium were incubated at 37 oC during 24 h in both aerobic and anaerobic conditions. In anaerobic conditions the medium were placed in an anaerobic jar (BD BBL™ GasPak™), with a CO2 gas generators envelope (BD BBL™) and another envelope of BD BBL™ GasPak™ anaerobic indicator were placed on them.
Round 2
Reviewer 3 Report
Thank you for improving the manuscript.
